# Applications of Smart Material Sensors and Soft Electronics in Healthcare Wearables for Better User Compliance

**DOI:** 10.3390/mi14010121

**Published:** 2022-12-31

**Authors:** Arnab Ghosh, Sagnik Nag, Alyssa Gomes, Apurva Gosavi, Gauri Ghule, Aniket Kundu, Buddhadev Purohit, Rohit Srivastava

**Affiliations:** 1Department of Biosciences and Bioengineering, Indian Institute of Technology Bombay, Powai, Mumbai 400076, India; 2Department of Biotechnology, School of Biosciences & Technology, Vellore Institute of Technology (VIT), Tiruvalam Road, Vellore 632014, Tamil Nadu, India; 3DTU Bioengineering, Technical University of Denmark, Søltofts Plads 221, 2800 Kongens Lyngby, Denmark

**Keywords:** smart materials, diagnostics, nanotechnology, healthcare

## Abstract

The need for innovation in the healthcare sector is essential to meet the demand of a rapidly growing population and the advent of progressive chronic ailments. Over the last decade, real-time monitoring of health conditions has been prioritized for accurate clinical diagnosis and access to accelerated treatment options. Therefore, the demand for wearable biosensing modules for preventive and monitoring purposes has been increasing over the last decade. Application of machine learning, big data analysis, neural networks, and artificial intelligence for precision and various power-saving approaches are used to increase the reliability and acceptance of smart wearables. However, user compliance and ergonomics are key areas that need focus to make the wearables mainstream. Much can be achieved through the incorporation of smart materials and soft electronics. Though skin-friendly wearable devices have been highlighted recently for their multifunctional abilities, a detailed discussion on the integration of smart materials for higher user compliance is still missing. In this review, we have discussed the principles and applications of sustainable smart material sensors and soft electronics for better ergonomics and increased user compliance in various healthcare devices. Moreover, the importance of nanomaterials and nanotechnology is discussed in the development of smart wearables.

## 1. Introduction

The practice of medicine has increasingly become technology-dependent along with user adaptation to smart devices. However, adherence and compliance remain two major issues that can fail even a technically superior invention. The efficiency of a biomedical invention improves if these factors are considered during the developmental cycle of a product. Particularly, the manufacturers developing wearable biomedical sensors to monitor the various vital signs and diseases in patients should be conscious of the patient’s needs and a physician’s approach to treatment. 

Adherence vs. Compliance: In clinical practice, often the terms compliance and adherence are used interchangeably. Adherence implies that a patient agrees with the provided advice, whereas compliance implies that the patient is passive. In both cases, a “physician’s control is exaggerated over the process of administering medications”, as described by Steiner and Earnest [1]. These behaviors can affect the treatment outcomes of any disease. When it comes to chronic diseases, many factors contribute to adherence and compliance to therapies. Usually, the sole blame is imposed on the patient in case of treatment failures. However, many factors can hinder habit formation, for example improper fitting of braces, pain or discomfort arising from accessories, cosmetic issues, heating up of devices, noise, obnoxious stimulus, annoying sounds emitted from the device, irritation, inaccessibility to servicing, charging issues and many more. The complexity of using various electromechanical devices needs to be identified from the perspective of ergonomics, user comfort, interfacing, conformity, and customization. This has shown to improve compliance and adherence to therapies.

In the traditional method of parametric follow-up, like monitoring glycemic control, a patient visits a pathology lab to have blood drawn and checked periodically as advised by the treating doctor. Sometimes this is logistically inconvenient. The results of the tests can take up to two to three days in many cases. Most likely about 40–50% of patients do not take prescribed drugs correctly (as, for example, inhaler use in case of bronchial asthma treatment) [2]. Furthermore, this number is comparatively higher in developing countries [3]. Non-adherence behaviors, on the other hand, include a diverse range of things, from irregular drug intake and short periods of not taking medicine, to giving up on treatment too soon [4]. Poor adherence to treatment, on the other hand, can come in many forms, for example, trouble in operating a device, maintaining frequency, and following the instructions sequentially. These factors show up as errors in omission, dose, time, or purpose (misuse of one or more medicines). Not going to appointments on the day and at the specified time and not making the necessary changes to habits and lifestyles are also adherence issues that can arise due to poor compliance. 

Compliance tracking: Patient compliance with drug dosage regimens can be tracked using a variety of techniques by physician and pharmaceutical companies [1]:Self-reporting by patients in the form of follow-up visits, questionnaires, or disease reports;Pill count is a common method of estimating compliance by comparing the number of pills consumed to the number of pills dispensed;Preventative measures such as monitoring vital signs such as blood pressure and glucose levels.

Smart Materials based wearable healthcare devices:

There is no consensus on how to define “smart materials”. For example, NASA describes them as “materials that remember different forms and are able to reconcile with particular stimuli”. It can also be described as highly-engineered materials that can react smartly to their environment. Smart materials belong to the category of advance materials which can sense (sensor component) and act/respond accordingly (actuator component). Some unique qualities of smart materials are as follows: Transiency: they are sensitive to their environments and capable of existing in a number of states;Immediacy: swiftly reacting to environmental influences;Self-actuation (intelligence): this ability is contained within the matter;Selectivity: the reaction is distinct and predictable;Directness: co-occurrence of action and reaction.

Some of the most important class of smart materials used in sensors (Table 1):

Smart sensors can be used to monitor patient compliance using smart blisters and smart bottles [5,6]. Smart pill blisters monitor a patient’s intake using electronic circuits. Smartphones can then use near-field communication (NFC) or Bluetooth to transmit this data to a digital platform. A digitalized adherence method allows for the user registration and user verification of medication intake via sensor-enabled medication packages (smart packages) to eliminate the guesswork in medication intake. Both patient convenience and data integrity are enhanced by this method [7]. With smartphones and smart devices, it is possible to incorporate additional features such as capturing surveys, sharing information via multimedia content (e.g., electronic labels, electronic consent), and capturing data from wearable devices. In addition, the electronic adherence methods enable the site to monitor patient adherence patterns and make data-driven decisions. There are numerous methods for preventing patients from dropping out, including proactive engagement and the implementation of mitigation measures [8]. Patients and biopharmaceutical companies have a reasonable incentive to work together in discovering, developing, advancing, and providing newer treatment options. A patient-centered approach is made possible partly by technology, which can play a critical role [9]. Smart materials play a significant role in healthcare due to their ability to respond to external stimuli such as light, temperature, moisture, stress, pH, and electric or magnetic fields. Arguably, the most crucial feature of these devices is their ability to revert to their original form as soon as the stimulus is removed [10]. The design and properties of these devices, along with their insight, have prompted their popularity as “smart materials” that are sensitive enough to detect very small changes [11]. 

Smart devices can be used for adherence measurements due to their precision and can eliminate tedious processes which make it easier for the patient to adhere to complex study procedures. The recent developments in smart materials with high flexibility and compatibility with the skin can provide optimum specificity and selectivity for developing a wearable sensor. Biocompatibility, durability, sensitivity, selectivity, form factor, stretchability, washability, and self-healing properties have all been taken into consideration in the designing of these devices, which have been made from a variety of rigid, stiff, and flexible materials so far [12,13]. Smart materials have been incorporated into various devices, such as stretchable electronic skins, micro-machined semiconductor reservoirs for electronic capsules, and drug delivery patches for the skin with polymer-based micro-needles and bladder actuation with SMA (Shape Memory Alloy) based actuators [14,15]. Materials with high stretchability and flexibility to fit over the curvilinear human skin, and flexible to allow bending, twisting, and stretching are used in the healthcare wearables. Ceramics, polymers (such as polyimide (PI), polyurethane (PU), polyethylene naphthalene (PEN), polyethylene terephthalate (PET)), elastomers (such as polydimethylsiloxane (PDMS)), and polymers with hydrogel properties are some of the common materials used [7,16]. A potential benefit of these materials as active and intelligent structural materials is their ability to respond to external stimuli (e.g., electrical, mechanical, chemical, or magnetic signals).

The latest wearables have incorporated the unique property of energy harvesting, making the sensor battery-free and self-sustainable. This increased the scope of the application of sensors for several conditions and would help make the wearables accessible in telemedicine. These sensors could be useful in rural and remote areas where there is still lack of electricity. In the future, nanosystems designed to be self-powered for other integrated transducers will open up a new field of research. A broad range of sensing and actuation applications in healthcare can be achieved through smart materials because of their ability to respond to external stimuli like changes in temperature, stress levels, pH, light, moisture, and electric or magnetic fields [12,16].

This paper talks about the state-of-the-art smart materials and soft electronics-based micro/nanosystems proposed for diagnostic, therapeutic, and treatment applications that can be implanted or worn. Apart from the flexibility or adaptability of the transducers (sensors and actuators) and the uniqueness of the materials used to build them, a unique thing about these systems is that they are built with different types of energy harvesters that allow the whole thing to run without batteries. When these self-sufficient systems are integrated with information technology, it has big effects on smart healthcare [9].

## 2. Healthcare Wearables

“Wearable devices” are tools that can be worn directly on different parts of the body. These gadgets are popular because they make it easy to continuously get important information about a person’s health in real-time and most of them operate non-invasively. Wearable device-based non-invasive or minimally invasive diagnosis opens up new possibilities for remote and continuous healthcare monitoring in non-clinical settings, with the ability to spot developing diseases between regular check-ups. Wearable medical technology also makes it easy and affordable for people to be more involved in their health care, which increases compliance.

Numerous monitoring systems for parameters, like heart rate, respiration rate, and temperature are well-known. Recently, non-invasive biomarkers in saliva, sweat, tears, and other body fluids are more in focus for the diagnosis of many diseases. As wearable devices now becoming more compact and portable, it is also changing the way how healthcare professionals interact with patients, conduct tests, collect data, and administer medications [17]. 

Wearable sensors are becoming a part of primary source of health information, for example, smart watches, neckbands, headbands, eyeglasses, shirts, and shoes are becoming more popular to keep track of health parameters. Most of them are equipped with sensors that gather raw data, which is then sent to a database or software application for processing. A response is frequently the result of analysis, such as notifying a doctor to speak with a patient experiencing unusual symptoms or congratulating someone on reaching their fitness or dietary objectives [18]. This uses theory, concepts, information, and techniques to enhance system functionality and human well-being. Understanding interactions between people and other system components is a matter of further discussion. This field in particular promotes a human-centered, holistic approach that considers relevant environmental, organizational, cognitive, physical, and other factors. Present-day data registration by using smart sensor systems in wearables like insole pressure systems, or body-mounted smartphones, are the strength for future AI-enabled auto-diagnosis and precision technologies. It is simpler to keep track of the assessment parameters and locate the most urgent issues through graphic interfaces, such as GUIs [19]. To ensure an effective identification of the most crucial parameters, work should be put into developing user interfaces that offer users a variety of pertinent information and can be easily adopted without requiring programming knowledge and skills. There are some excellent review articles that have discussed the properties and sensing mechanism of commercially available wearable healthcare devices [20,21,22,23]. 

Smart wearable devices have proven to be very popular in the entertainment, information, communication, medicine, health, and fitness industries. The health and fitness sector remains a key market driver, and fitness trackers remain the most popular product category by unit sales. Smart wearable technology is now easily incorporated into many aspects of our daily lives, and the range of products available to consumers is constantly growing [24]. As the functional capabilities of wearable devices increase, risk assessment and forecasting should be enhanced [25]. Regulations frequently require explicit or oblique adherence to the necessary privacy and data security standards [18], for instance, by taking special measures to guarantee that the data is only accessible to the users of the smart wearables themselves or their healthcare providers.

User-centric monitoring is as follows: the main advantage of wearable technology is its user-centric approach to data collection and data processing. For example, people with diabetes had to go to their doctors to check their blood sugar levels, whereas today, they can check their glucose level with a simple hand-held device, or a sensor implanted in their arm [26]. Some of the frequently used non-invasive wearable medical equipment are listed below in Table 2.

## 3. Smart Materials and Soft Electronics in Wearables Healthcare Devices

### 3.1. Temperature Sensors

Body temperature is a significant vital sign that reflects physiological well-being and acts as a manifestation of inflammation inside the body. Body temperature monitoring makes it possible to comprehend problems with human health such as cardiovascular health, wound healing, pulmonological diagnostics, and syndrome prediction. In older methods, paste-on temperature sensors or infrared digital cameras have been used. These techniques might have some benefits but cannot provide cost-effective, continuous temperature mapping [28]. Today, various nanomaterials are used to create wearable temperature sensors. More specifically, graphene, conductive polymers, carbon nanotubes (CNTs), nickel, and copper metal nanoparticles have been used to create thermal sensing components [29]. Flexible wearable sensors need a flexible substrate to stabilize the active material. In this regard, polyurethane, PDMS, polyethylene terephthalate, eco flex, and polyethylene naphthalate are the flexible substrates most frequently used in wearable sensors for health monitoring [43].

Liu et al. reported a reduced graphene oxide (r-GO)-based temperature sensor with one possible application for internet-of-things (IoT) and electronic skin [44]. They tested r-GO, single-walled carbon nanotubes (SWCNTs), and multi-wall carbon nanotubes (MWCNTs) for the temperature sensor as a function of the change in resistance in the range of 30 °C to 100 °C. r-GO was found to be the most sensitive material for temperature sensor development with a very high degree of insensitivity towards surrounding environments like humidity or other gases. Figure 1A shows (a) the fabrication steps of the temperature sensor on a flexible PET layer, (b) the final sensor, and (c) the subsequent application of the sensor on a robotic arm. For its exceptional thermal, electrical, and mechanical properties, graphene and its derivatives have been used in the development of numerous wearable temperature sensors [45]. Wu et al. have reported the development of a conductive hydrogel-based thermal sensor with a wide working range of −28 to 95.3 °C [46]. Based on cross-linked poly(3,4-ethylene dioxythiophene): poly(styrene sulfonate) (PEDOT: PSS), Wang et al. developed a fully printed flexible temperature sensor with high stability in environmental humidity that varied from 30 percent RH to 80 percent RH and a high sensitivity of −0.77 percent °C for temperature sensing between 25 °C and 50 °C [47]. The ability to wirelessly sense temperature is made possible by integrating the printed sensor with a printed flexible hybrid circuit.

### 3.2. Strain Sensors

Flexible, skin-mountable, wearable strain sensors are required for various applications, including personalized health monitoring, human motion detection, human–machine interfaces, soft robotics, and others. In addition to their high efficiency, they must also meet the minimum standards, which include high stretchability, flexibility, durability, low power consumption, biocompatibility, and lightweight. These requirements are even stricter for epidermal electronic devices, which must have mechanical compliance similar to human skin and high stretchability (strain > 100%) [48]. Stretchable strain sensors have been created using low-dimensional carbons like carbon black, carbon nanotubes, and graphene, as well as nanowires, nanoparticles, and hybrid micro/nanostructures [49,50,51,52]. Rubbers (such as natural rubber and thermoplastic elastomers) and silicone-based elastomers (such as polydimethylsiloxane (PDMS), Ecoflex, and Dragon Skin) are the most frequently used polymers as flexible support materials for soft strain sensors (TPEs) [53]. Seyedin et al. reported the development of a Ti3C2Tx MXene/polyurethane (PU) composite-based fibers with high stretchability and conductivity for textile-based wearable strain sensor development [54]. The same group had also reported the development of cellulose-based fabrics modified with MXene for strain sensing along with energy harvest and energy storage [55]. Shintake et al. have reported another type of wearable strain sensor from Carbon Black/elastomer composites and its application on a smart glove [56]. Figure 1B shows the design of the sensor (a), and the development of five different strain sensors on a single layer, one each for a finger (b). The movement of the fingers (equipped with a different sensor) of the gloves can be monitored using the capacitance signal output (c), which can be used in future for medical uses.

### 3.3. Detection of Sweat Metabolites

Many wearable sweat sensors have been recently created, each with a different form factor, substrate, and sensing mechanism. To track fitness over time, popular athletic accessories with built-in sensors, such as wristbands or headbands, can be worn comfortably and without impeding movement. Patch-style formats are frequently used in medical applications because they offer greater location flexibility and can covertly attach to the skin. Numerous substrates, including temporary tattoos, soft polymers, and hybrid systems that combine flexible plastics with conventional silicon integrated circuits, can produce these form factors (ICs). Numerous sensing techniques, including electrochemical, colorimetric, optical, and impedance-based detection, can be used to find analytes in sweat [57]. Hartel et al. have reported the development of an electrochromic material-based resettable biofuel-based sensor for sweat lactate monitoring [58]. Figure 2 shows the design of the sensor (A and B) with a lactate-oxidizable biofuel cells (BFC) anode, oxygen-reducible BFC cathode, and a visual display of reversible Prussian blue (PB) electrochromic material, all printed onto a single wearable patch. This sensor can be used in various medical settings for visual detection of sweat biomarkers. Gao et al. reported the first fully integrated wearable sensor system for multiple analyte detection in the sweat samples [59]. Following this report, in the next five to six years, the sweat sensing sensors have seen a tremendous growth in technologies and applications, however most of them are limited to the detection of ions and metabolites. The detection of more complex protein or other biomolecules is still a challenge to the current technology.

### 3.4. Detection of Volatile Biomarkers

Volatile organic compounds (VOCs) can be used for health monitoring in POC devices using non-invasive methods. Structurally different hydrocarbons, primary and secondary alcohols, aldehydes, ketones, esters, nitriles, and aromatic compounds are secreted from breath, skin, feces, saliva, urine, and blood [60]. VOCs in breath, sweat and saliva have been used for monitoring pathophysiological conditions [61]. E-noses is a success story of the development of sensors for VOCs detection, as it is reported to detect various diseases such as lung cancer and diabetes [62]. However, the major limitation of VOCs sensors is the lack of selectivity towards the target analyte leading to false positive results. Most of the recently reported VOCs sensors therefore use nanomaterial modified sensor matrix to enhance the sensitivity of the sensor, and IoT and artificial intelligence to improve the selective detection of the targets through pattern recognition. 

Vishinkin et al. have developed a volatile compound collection method using a polymer, poly(2,6-diphenylphenylene oxide) polymer, and a machine learning integrated nanostructured sensor for the detection tuberculosis on skin [63]. Detection of TB is a unique case, as instead of one, there are multiple volatile biomarkers from skin which are involved here. The sensor was able to distinguish between active pulmonary TB patients from the healthy control group with 89.4% accuracy. The use of such sensors for infectious diseases through VOCs detection is going to play an important role in future healthcare sectors. The development of a VOCs sensor follows confirmation of VOCs in the selected clinical sample through GC-MS or other mass spectrometry, development of nanoparticle modified sensor for sensitivity, development of pattern recognition algorithms using IoT and artificial intelligence, and development of wearable modules of the VOCs detection (as shown in Figure 3). 

### 3.5. Wound Health Assessment

Researchers have long been interested in dressings for wounds that hasten to heal. Size, type, internal and external wound environments, body temperature, body oxygenation, wound hydration, and infection are some of the characteristics of wounds. These factors are assessed for monitoring wound healing. The wound dressing is usually opened to monitor the wound healing process/efficacy of the treatment, which cause secondary injury, pain and additionally interrupts the normal wound healing process. The wound dressing can be integrated with smart sensors to monitor the healing process without changing the wound dressing frequently. The sensor can also provide real-time information on when to change the dressing or whether any wound dressing is required. Most of the reported wearable wound sensors are based on pH monitoring as the pH of healthy skin is different than different stages of wound. The pH of healthy skin ranges from 4 to 6, whereas the pH of chronic wounds ranges in between pH 7 and 9 [64]. For optical monitoring of the wound healing process, pH indicator dyes are used to distinguish between a healthy skin pH from a more alkaline pH [65]. Liu et al. reported the use of alginate/polyacrylamide hydrogel incorporated Phenol red dye for wound pH monitoring [66]. Similarly, electrochemical wearable devices based on different polymers, metal oxides and nanomaterials are used for wound monitoring. Guinovart et al. have reported a PANI-based electrochemical sensor screen printed on a bandage for pH monitoring in the range of 5.5–8 for wound health monitoring [67]. These sensors can be integrated with stimuli-responsive materials and self-healing materials for effective wound healing and monitoring [68]. 

### 3.6. Orthopedic and Surgical Site Assessment

Following spine surgery, wearable medical technology that tracks a patient’s mobility may help surgeons with post-operative care, treatment outcome evaluation, and patient mobility. Medical professionals must assess those devices using definite criteria and sub-criteria [69]. Assessing the general activity and performing a simple gait analysis are frequently the main areas of concern for surgeons, according to a study by Braun et al. that showed measurement intervals ranging from days to months [70]. The fact that many processes for evaluating outcomes involve determining joint angles and range of motion is another reason for the respondent’s value kinematic analysis. These steps enable more thorough examinations, including complete musculoskeletal simulations. Table 3 contains some examples of recent developments in nanomaterial-based sensors for healthcare applications.

## 4. Scope for Improvement in Ergonomics and User Compliance

Some of the problems that need to be solved include simultaneously monitoring various physiological parameters and improving the ergonomic design following user compatibility [46]. Research on hybrid wearable sensors, techniques for attaching and detaching sensor components, and thinner, softer, invisible electronics are some of the technology’s prospects [47]. It has been predicted that augmented/virtual reality, IoT, and increased human–machine interactions using various gestures will all be combined. The top three drawbacks of using smart electronics for healthcare purposes is depicted in Figure 4.

## 5. Recent Advances in Soft Electronics-Based Healthcare Monitoring

Flexible, self-healing, self-monitoring, and self-sustaining materials are some of the most exciting developments in smart materials. These qualities are crucial in enhancing the functionality, dependability, and overall user experience for various biomedical applications, such as tissue engineering and device implants [48,49], health/rehabilitation monitoring sensors [74,75,76], electronic skin [77,78,79], and textile-based healthcare wearables [80,81]. Research in this field has also benefited from 3D and 4D printing technology improvements in the capacity to print biomaterials [82]. The effectiveness and dependability of sensor-based systems have increased thanks to integration with associative technologies like artificial intelligence (AI), machine learning (ML), and IoT, particularly in the areas of data collection, data handling, and real-time monitoring [83]. The Internet of medical things (IoMT) integrated a biosensor patch for remote monitoring of vital signs [84], allowing for remote monitoring of vital signs [85,86,87]. Developing self-sustaining sensors using nanotechnology has also received significant research attention. 

Development of flexible smart materials: without losing their integrity, flexible materials can be stretched, compressed, and twisted into different shapes [88]. To achieve these qualities from a structural standpoint, designs like buckled nanoribbons, serpentine-interconnect, and island-interconnect designs have been created [89]. Apart from these, Japanese techniques like origami and kirigami have been used as inspiration to alter the behavior of materials. While origami solely involves paper folding to generate complex shapes, kirigami involves paper folding and cutting. Some examples based on these techniques include smart adhesion systems [90], a self-foldable chemical sensor [91], shape memory polymers and composites [92,93], and a bifurcated stent [94].

Self-repairing: self-healing materials can automatically repair themselves after suffering any damage from the external environment. The healing capabilities of such materials can either be intrinsic or extrinsic [95]. The intrinsic healing process is based on reversible molecular bonds in the material, which may require an external stimulus such as heat, pressure, or light to catalyze the healing process. An example is the bio-inspired self-healing electronic skin developed by Cao et al., which is based on dipole-ion interactions [96]. On the other hand, in extrinsic materials like the polymer composite developed by White et al., an autonomous microencapsulated healing agent triggers the self-heal process based on external damage to the area. However, the integrity of the healed area is usually inferior to the original material, and the material loses its self-healing capabilities in the damaged area over time as the healing agent is depleted [97]. Techniques like microvascular coating networks [74] have been investigated to combat such limitations. A thorough focus in this area is required to improve the mechanical and electrical properties of the material post repair to produce robust materials for wear-intensive applications. 

Energy harvesting: the main stumbling block in healthcare technology is the ability to power medical devices, particularly implantable ones. Devices to harvest biomechanical energy of the body have been developed to make implantable micro/nano self-powered systems that can operate long without needing to replace batteries. Various energy harvesters (EHs), such as triboelectric nanogenerators (TENGs), thermoelectric generators (TEGs), and piezoelectric nanogenerators (PENGs), malleable solar cells and photovoltaic (PV) platforms, for sunlight’s energy harvesting and light converting biochemical react to an electrical signal by using chemical and gas sensors [98]. These materials are also an excellent choice for harvesting biomechanical energies from body motions, the external environment, or body heat. Hence, smart wearables are a reliable and effective way to monitor patient vitals. It uses smart materials to provide a sensing platform, and electronics help collect and process data [99].

IoT: with the evolution of the Internet of Things, small and flexible devices can now be built into people’s bodies. This would change how we interact with the outside world and improve our lives. These are called smart wearable or mobile health technology devices. Electronics that can be bent or stretched are used in many wearable devices, like watches, glasses, clothing, and even patches that look like tattoos. Health technology can benefit from a wide range of new materials and designs and from combining data from different smart sensors using IoT [100,101,102]. Prosthetics and robots, health care, biomedical systems, and tracking fitness are a few ways this technology can be used [15]. 

Microneedles (MN)-based electrochemical (bio)sensing is becoming an emerging field of research and development due to its potential to continuously monitor clinically important molecules in non-invasive samples [103]. MN-based sensors are popular for theranostic purposes because they can be used directly to analyze interstitial fluid (ISF), which is a rich source of information about biologically relevant molecules and biomarkers. MN-based sensors can avoid some of the major problems with blood analysis, such as painful extraction, high concentrations of interfering molecules, and incompatibility with dialysis (especially in newborns). The majority of MN devices are still in the in vitro and ex vivo stage of development demonstrating adequate performances through phantom gel and/or sectioned animal skin, with only a small number having been used to demonstrate real in vivo measurements in animals and humans [104,105]. Electrochemical MN (bio)sensors have traditionally focused on glucose [106,107], but interest in other species, such as lactate, drug compounds, nitric oxide, pH (hydrogen ions), and others, is growing [108]. Therefore, in the search for more stable and reliable sensors, there has been a noticeable shift in the reported sensors away from amperometric enzymatic detection and toward non-enzymatic approaches. Each method of implementing the electrochemical electrodes—using a single or several parallel networks of MNs—has its own benefits. Multi-analyte detection using MN devices, such as the simultaneous detection of glucose and lactate, has also been proposed in a number of works [109]. The shortcomings in the protocols used for the calibration and validation of electrochemical MN (bio)sensors, as well as a lack of biocompatibility (cytotoxicity and inflammation) and durability studies, in relation to the assessment of the analytical features of these sensors, are some limiting factors for in vivo applications of MN-based sensors. These challenges, along with those involved in obtaining the necessary formal approvals for in vivo studies, are likely to be a stumbling block for electrochemical MN (bio) sensing technology as it seeks to enter the commercial market. MN-based biosensors can be integrated with drug delivery modules for the efficient sensing of biomarkers as well as disease management [105,106].

## 6. Nanotechnology for Smart Materials and Soft Electronics

Nanotechnology has revolutionized research across different domains in the 21st century. In the healthcare industry, nano-based systems have contributed to novel therapeutics, diagnostics, and treatments in the form of wearable and implantable devices and interventions [11]. These approaches have effectively dealt with age-old issues in biomedical products like biocompatibility, sensitivity, reliability, multi-functionality, user compliance, and comfort. Some of these advances are listed in Table 4. 

One of the biggest problems in smart material sensors and devices is providing an efficient and long-lasting power supply while restricting the weight and bulk of the device. Often, researchers are forced to come to an impasse at the expense of user compliance and comfort. This dearth gave rise to an era where energy is harvested from the wearer’s environment or the human body itself [11]. Presently, energy harvesting systems are of the following types: solar or biofuel cells, triboelectric, or thermoelectric nanogenerators [11,116]. Examples of these include cardiac motion-powered pacemakers [117], sweat-activated batteries [116], radiofrequency-powered subcutaneous optoelectronic systems for neuroscience research [118], piezoelectric knee-joint movement-based energy harvesters [119], and wave-shaped piezoelectric composite for blood-pressure based energy harvesting [120]. Figure 5 highlights the present scenario of different healthcare devices and sensors.

## 7. Challenges and Future Perspectives of Smart Materials for Wearable Healthcare Devices

Smart devices are increasingly being used for diagnostics, monitoring, therapeutics, drug delivery, rehabilitation, and surgery. Despite a multitude of applications, they have shortcomings that need critical attention for better user compliance and optimal performance. 

Wearable temperature sensors: These are designed to monitor the thermal conditions of the skin continuously. Temperature sensors are either thermally resistant or thermally sensitive field-effect transistors (FET) and work at temperatures ranging between 35–42 °C. Thermal resistance-based temperature devices are metal-based devices whose sensing range is limited at high temperatures, i.e., >42 °C, due to low thermal response [121]. On the other hand, FET-based temperature devices have greater sensitivity with high accuracy in determining skin temperature. However, an improper response to thermal changes, inadequate stability, and significant interference from the ambient environment are some of the challenges all conventional wearable temperature devices face. Hence, it is necessary to develop more sophisticated electronic devices. Graphene nanowalls combined with polydimethylsiloxane are incorporated in a thermal resistance sensor that works in temperatures ranging between 35–45 °C, sufficient enough to monitor human body temperature. A 15 μm graphite–polymer-based ultrathin thermometer of 15 μm has been developed to precisely measure temperatures between 25–50 °C [122].

Wearable strain sensors: These are mostly used to detect signals such as the heartbeat and respiration rate. Strain sensors should be designed to be lightweight, reliable, flexible, and stretchable to match the mechanical properties of the skin. Generally, their working mechanism is based on piezoresistive, piezocapacitive, or piezoelectric principles [123]. Piezoresistive wearable strain sensors are very sensitive and are made from electrically conductive flexible materials. If these are bent or deformed, there will be changes in electrical resistance and loss of electrical connection. Moreover, since these films are thin and brittle, they are prone to cracks that compromise their sensitivity. In response to these limitations, ultrathin gold nanowires, piezoresistive conductive polymers, and their composites have been developed for optimum flexibility and biocompatibility [124].

Wearable devices to record electrical conductivity through the skin: Such devices help monitor the electrical conductivity of the heart to detect an arrhythmia and decide whether the patient needs medicine, a pacemaker, an implantable cardiac defibrillator, or surgery. The sensor uses disposable silver/silver chloride-gelled electrodes. The main disadvantage of the electrode is its limited storage time (i.e., less than one year) and its non-reusability. Gelled electrodes are also less biocompatible and can sometimes cause irritation or redness when coming in contact with human skin [125]. However, recently-developed dry electrodes made of polymeric and textile-based materials are quite comfortable when in contact with skin and have long-term stability [126]. 

Wearable sensors for sweat metabolites: This non-invasive device is used to analyze the levels of sweat metabolites and electrolytes which are essential in distinguishing between the healthy and diseased states of the body [127]. For optimal functioning, keeping the device in close contact with the body is necessary. The base material is made using fabric/flexible plastics, which generally requires unique technology for its fabrication [128]. Screen-printed electrodes have many manufacturing issues depending on the properties of the material, its ink composition, etc., which would affect the performance of the device. Hence, there is a need to develop a noble catalyst for the inks used, to enable the specific functioning of the device. Moreover, the optimum annealing temperature needs to be identified to prevent electrode and substrate deformation [129]. These limitations are overcome using stamp-transferred electrodes and epidermal-suspended electrodes. The process is similar to tattooing, where electrodes are printed onto the skin. This method requires carbon fibres which are efficient in measuring the levels of different biomarkers [130].

Wearable sensors for volatile biomarkers: Metal oxides can detect volatile organic compounds depending on their chemical components, surface area, the microstructure of sensing layers, humidity, and temperature [131]. However, not all metal oxides possess the properties required to recognize the presence of biomarkers adequately. A significant challenge in developing such sensors is water adsorption which gradually lowers their sensitivity. The adsorption of water on the metal surface obstructs the movement of electrons to the sensing layers. On prolonged exposure to the environment, the oxide layer forms stable hydroxyl ions, resulting in progressive surface degradation. Therefore, proper temperature balance is required to eliminate humidity interference [132]. Conducting polymers are gas sensors that involve the adsorption of volatile compounds which separate the polymer chains, increasing electrical resistance. A hybridized form of such polymers can prevent this drawback [133].

## 8. Challenges in Wearable Healthcare Devices: The User Perspective

Data quality is an important aspect of scientific data that must be evaluated to ensure the validity of scientific claims. Data quality is also one of the fundamental values of research ethics and one of the social objectives of biomedical research; high-quality data are regarded as the foundation for clinical and extra-clinical benefits. Nonetheless, the heterogeneity of sensors and inconsistency of data collection in the context of wearables makes it difficult to coordinate and evaluate quality. Furthermore, the lack of contextual information regarding the collection, classification, and interpretation of wearable data raises concerns about the possibility of assessing quality. However, issues affecting balanced estimations in screening and prediction call into question the validity and foundation of wearables as detection and prediction tools. Several models have been used in the COVID-19 pandemic to forecast how the disease will evolve, spread, and impact, but they have also drawn criticism for their hazy underlying premises and finite scope [134,135]. Overestimation has a significant impact on wearable COVID-19 detection applications. Because elevated heart rate can be interpreted as a symptom of respiratory illness in general, it is frequently difficult to distinguish between COVID-19 and seasonal influenza and cases of standard influenza using wearable data [136,137]. As a result, wearables frequently detected and predicted COVID-19 infections incorrectly. Aside from these, health equality and skewed information are concerns for the predictive analysis of wearable data. The major concerns with wearable devices are mentioned in Table 5.

## 9. Future Scope

In essence, advancements in the creation of wearable medical devices are accelerated by the tremendous development in the field of material science and engineering. Our approach to disease diagnosis has recently changed due to novel sensing techniques. Real-time, continuous monitoring devices can give a better assessment of the patient’s health status and enhance the patient’s quality of life, and this expectation is unquestionably reasonable. Wearable healthcare devices offer cost-effective solutions, remote monitoring, and will be helpful in indicating an efficient treatment for early diagnosis. Many organizations are keen to explore this field of biosensors depending on the cooperation between material sciences and electronics, as well as data acquisition and signal processing, and need to make significant progress in meeting market demands [13]. The need for active sensing based on smart materials for wearable healthcare devices has greatly increased the interest in creating small-size NPs with single components and has prompted researchers to look for innovative design methods and producing sensing components possessing novel properties. In light of this objective, a deeper comprehension of sensing mechanisms can offer a clear understanding of the correlation between sensing properties and the composition, structure, and morphology of materials, which will help scientists and engineers as they investigate high-performance medical devices [138].

Even though various wearable sensors have produced encouraging results, significant barriers still stand in the way of the general public from using wearable technology regularly. First, few “bio-affinity” protocols are available [99] that describe how these biomarkers relate to a particular disease or physiological index. Even though this topic is covered in some reports, the connections at their core are still unclear. Second, device accuracy, long-term stability, sensitivity, and biocompatibility should all be enhanced to meet regulatory agencies’ standards for diagnostic devices. Third, the power consumption of the devices need to be effectively reduced [139].

Wearable sensors have an in-built sensing matrix which records body information which consumes a lot of power. Even though the creation of self-powered devices seems a possible solution to this problem, the effectiveness of their power generation is still in question. Although smart materials’ importance has been validated in parameters of designing high-performance, multifunctionality, compact skin-based wearables, still there is not enough progress in materials sciences related fields. Biomaterials have primarily been used for therapeutic purposes; in particular, they have demonstrated special qualities when applied to implantable medical technology [138]. It is anticipated that biomaterials-based skin-compatible wearable devices will achieve even more unexpected results because of their inherent biocompatibility and biodegradability. One potential advantage to switching to the regularly used polymeric elastomer for biocompatible biomaterials substrates, such as protein films. Their biodegradability will be particularly helpful in developing environment friendly products. Multifunctional materials, and the more common biomaterials, require more focus [13].

These multifunctional materials possess properties of spontaneous energy harvesting and multi-stimuli-responsiveness, as discussed in the section on multifunctional devices, will majorly reduce the cost, size dimension, and complexity of manufacturing these medical devices. Although there have not been many successful attempts at relevant multifunctional candidates up to this point, we predict that wearables made of intelligent multifunctional materials will spur additional creative developments. Overall, it is anticipated that a lot will fit into a patch owing to the developments in battery technology, materials sciences, sensor sophistication, and miniaturization [16]. There is a need for trade-off between the data entries collected, the amount of time used, and the processing power, just like with any wearable technology. In general, the accuracy of the data will increase with the proximity of the wearable device to the signal it is gathering. Patches can detect subtle changes in physiological condition of the patients and provide a thorough health profile of patient status because they allow direct skin contact [9].

## 10. Conclusions

Non-invasive diagnostics and user-friendly wearable materials are now possible owing to the rapid exponential developments in the field of material science. Disease diagnosis and prognosis fields have undergone a revolution due to the introduction of recent novel technologies and advancements. In recent years, healthcare wearables offering non-invasive real-time monitoring and comprehensive medical status reporting have replaced conventional diagnosis. Strenuous advancements should be made in this field to keep up with the market’s accelerating demands and lessen the undeniable burden on the healthcare industry, which can only be accomplished with the cooperation of material sciences, nano-theranostics, and electronic sciences. Researchers can gain deep insights into the relationship between detection sensitivities and wearable material characterization to deliver effective healthcare services if they have a broad perspective and understanding of sensor mechanisms and their diagnostic interventions. Continuous innovation, careful consideration of the materials used, and innovative fabrication methods, all contribute to the steady advancement in the field of medical devices. From the perspective of emerging concepts, this article provides a brief overview of some of the recent innovations and breakthroughs in the field of wearable technology.

## Figures and Tables

**Figure 1 micromachines-14-00121-f001:**
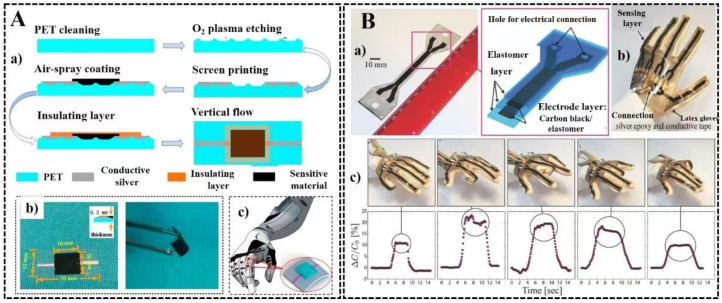
The advancement of soft electronics and smart materials-based sensors for temperature sensing and strain monitoring. (**A**) The development of a flexible PET and graphene oxide-based temperature sensor showing the (**a**) the fabrication steps of the sensor, (**b**) the final sensor, and (**c**) the application of the sensor for internet of things. Reproduced from Liu et al. under a Creative Commons Attribution 4.0 International License (CC BY 4.0). Copyright (2018); (**B**) Wearable strain sensor developed by using a carbon black-filled elastomer composite. (**a**) The different layers of the sensor, (**b**) five different strain sensors developed on a glove, and (**c**) the use of an intelligent glove fabricated with strain sensors. Reproduced with permission from Shintake et al. © 2022 Wiley.

**Figure 2 micromachines-14-00121-f002:**
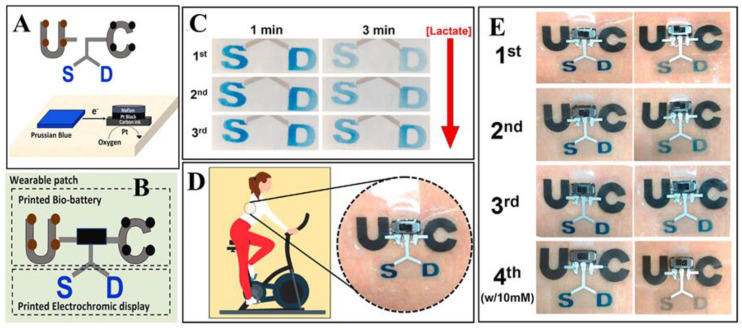
Sweat-powered wearable electrochromic biosensor for lactate monitoring. (**A**) Design and working principle of the wearable sweat sensor. (**B**) Different parts of the sensor i.e., printed bio-battery and electrochromic display. (**C**) Change in color of the electrochromic display after addition of human sweat samples. (**D**) Schematic of the sensor applied for on-body measurements, and (**E**) On-body images of the device with reversible behavior up to four switching cycles. Reproduced with permission from Hartel et al. © 2022 Elsevier.

**Figure 3 micromachines-14-00121-f003:**
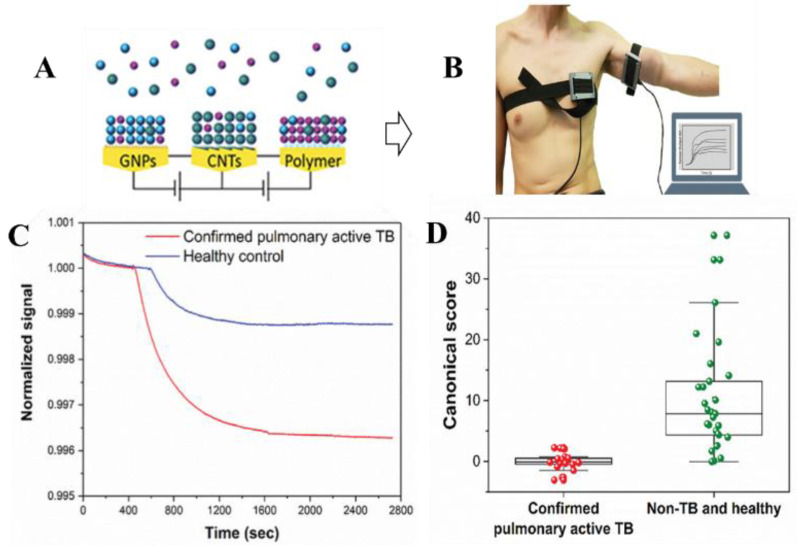
VOCs sensor for tuberculosis detection. (**A**) Nanoparticle modified sensor array; (**B**) Wearable devices on the chest and anterior arm of a volunteer for VOCs measurement; (**C**) Normalized signals of the wearable device attached to the anterior arm area; (**D**) Use of Discriminant Function Analysis (DFA) based pattern recognition to distinguish between healthy and diseased samples. Reproduced with permission from Vishinkin et al. © 2022 Wiley (under Creative Commons CC BY license).

**Figure 4 micromachines-14-00121-f004:**
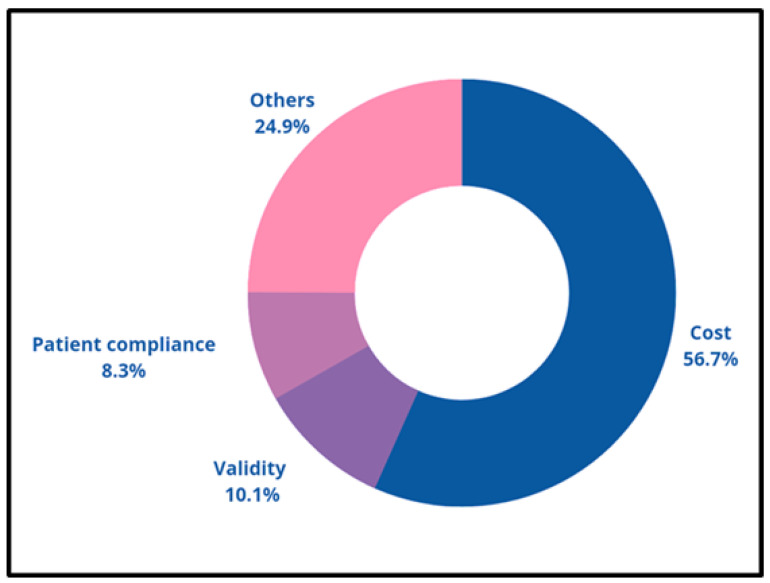
Diagrammatic representation for major limitations concerning the user compliance and ergonomics.

**Figure 5 micromachines-14-00121-f005:**
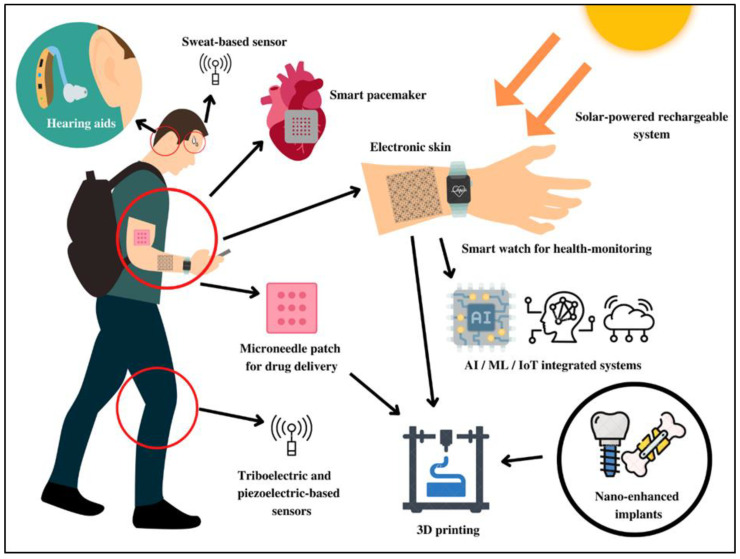
The multifaceted application of smart healthcare electronics fabricated with AI-ML/IoT integrated systems.

**Table 1 micromachines-14-00121-t001:** Classification, Mechanism and Use of Smart Materials.

Class	Mechanism	Use
(a) Shape memory materials	Shape memory materials recover to their original shape after a significant deformation due to some stimulus/stress.	Robotics and textile/fashion industries
(b) Piezoelectric materials	Generates a voltage when deformed or vis-à-vis	Electronics devices (transducers and sensors) Pressure sensors
(c) Electrostrictive materials	Like piezoelectric materials, but the mechanical change is proportional to the square of the electric field.	Similar to piezoelectric materials
(d) Magnetostrictive materials	When put in a magnetic field, or vice versa, they change and cause mechanical stress	Sensor and an actuator energy harvesting
(e) Rheological materials	These are liquids that can instantly change from one state to another when an electric or magnetic charge is applied.	These fluids may be used in car brakes, shock absorbers, and dampers.
(f) Thermo-responsive material	These materials change their properties in response to changes in temperature	Thermostats
(g) Electrochromic materials	These materials change their optical properties when a voltage is applied	Cathode in LCDs and in lithium batteries
(h) Biomimetic materials	Synthetic materials developed to replicate the properties of materials produced by living organisms	Nanozymes

**Table 2 micromachines-14-00121-t002:** Medical wearable devices currently used for health monitoring.

Sample Source	Physiological Parameter	Sensing Devices	Refs.
Daytime activities	Heart rate, blood pressure, and abnormalities	Wearable vests	[27]
Sleep monitoring	Heart rate, body temperature, and breathing	Smart rings and sleep mask	[28,29,30,31,32]
All activities	Heart rate, ECG, and level of oxygen in the blood	Earphones	[33,34]
Physical activities	Respiratory rate and level of oxygen in the blood	Smartwatches and wristbands	[35]
Blood monitoring	Thoracic impedance and various biomarkers	Arm-implants	[17]
Movement and rest, pressure	Body temperature, heart rate, EEG, ECG, and respiration rate	Textile-based devices	[36,37,38]
Sweat	Sweat rate measurement	Textile-based devices	[39]
Chest, forearm, and forehead, blood pressure	ECG, EEG, EMG, and arterial pulse	Tattoo-based or E-skin	[40,41]
The epidermis of skin in the arm	Epidermal signals and ECG signals	Piezoelectric patch	[42]

**Table 3 micromachines-14-00121-t003:** Recent advances in sensing materials in healthcare applications.

Application	Sensing Materials	Characteristics	Ref.
Temperature Sensor	PEDOT:PSS	High stability High sensitivity	[47]
Strain Sensors	Gold nanowire films doped with Polyaniline microparticles	Enhancement in stretchability, sensitivity, and conductivity Water resistible Durable	[53]
Sweat metabolites detection	3-dimensional lab-on-chip Au-based nanostructure	Detects cortisol in sweat Stretchable Label-free Highly sensitive	[71]
Wound healing	Chitosan-Tetrabenzaldehyde-Functionalized Pentaerythritol hydrogel	Injectable Biocompatible Self-healable Conductive	[72]
Volatile Biomarkers detection	Porphyrin/rGO/polyimide film	Simultaneous detection of physiological signals such as pulse rate, breath rate, and VOC such as ammonia, etc. Highly sensitive	[73]

**Table 4 micromachines-14-00121-t004:** Nanotechnology-based advances in smart materials and soft electronics.

Device/ Intervention	Nanomaterials Used	Applications	Ref.
Electrical bio-adhesive interface	Graphene nanocomposite	Rapid integration with wet tissues, unimpeded bidirectional communication at the device-tissue interface	[110]
Mouldable conductive nanocomposite	Silver nanowire networks with a stacked polymer structure	Integration with wearable epidermal electronics improves flexibility, breathability, and electromechanical stability	[111]
Gas permeable On skin electronics	Porous graphene and silicone elastomer sponges	Accelerates perspiration evaporation, minimizes the risk of inflammation, and contributes to user comfort	[112]
Bioresorbable electronic stent	Gold nanorod-silica nanoparticles incorporated with drug encapsulated nanoparticles	Effective flow and temperature monitoring	[113]
Implantable sensors for long-term monitoring of body fluids	Photostable gold nanoparticle-based nano platforms	Disease progression monitoring and therapeutic efficacy via biomarker concentrations	[114]
Self-powered metamaterial spinal fusion cage implants	Triboelectric auxetic microstructures	Condition monitoring for bone healing progress	[115]
Lab-on-a-patch non-invasive sweat biosensor	A microfluidics-integrated 3D nanostructured gold electrode	Immuno-detection of cortisol present in sweat	[71]
Electrical bio-adhesive interface	Graphene nanocomposite	Rapid integration with wet tissues, unimpeded bidirectional communication at the device-tissue interface	[110]

**Table 5 micromachines-14-00121-t005:** Key issues with wearable healthcare devices.

Areas	Issues	Recommendations
Data acquisition	Data quality Sensor variance Data collection method without contextual, pre-morbidity information	Standardisation of sensors, data collection methods, and customisation as per individual user-profiles.
Balanced estimation	Over and under estimation	Interoperability
Health equality	Everyone cannot afford to wear and use the technology. Most of these require internet-based operations. Therefore, many people are outliers from these benefits.	Increasing accessibility to the device and the data for inclusive benefit.
Representability	A few users compared to a large population use same wearable sensor devices. Therefore, the data cannot be representative of the entire population.	Standardisation of parts and components, method, process, manufacturing, and data analytics can bring in more parity.
Battery issues	Long-lasting batteries, easy to recharge and replace batteries	Upgrading battery technology with features like wireless charging with size-compatible batteries in devices
Complexity of Use	Complex hardware, software, and user interface of device can make the device less user-compatible	Elevating the user interface with minimum customization settings
Excessive air traffic	Results in blocking airwaves and thus loss of data	Exceeded bandwidth
Media Device Fatigue	Feels like a burden carrying several devices though they serve multiple purposes	Development of ergonomic and user-friendly device designs
Fractured Proprietary Development	Limited use cases of developed devices	Develop device that serves multiple purposes and can be beneficial for several different types for patients
Data Transmission	Invasion of privacy and data theft or data leak issues	Ensure the reliability and trustworthiness of data and access to contextual data. Setting compliance standards
Potential Health Problems	Internal working mechanism of devices shouldn’t cause health problems to users e.g., radiation, etc.	Appropriate scientific research and device quality assurance reports
Device Safety Issues	Inappropriate functioning in wearables	Ensure the device safety abiding safety regulations provided by the standardised organizations
Mobile Apps	Use of only mobile apps can limit the data accuracy and sensitivity	Mobile apps complementing user experience and overall working wearables can be preferred
User Distrust	User data privacy issue	Good company data management
Slow Vendor Progress	Either outdated or superfluous modalities could slower the vendor progress	Up-to-date device models according to the user requirements
Negative User Experience	Can lead to bad marketing of the products	Balanced set of features

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
