# Peer review of "Applications of Smart Material Sensors and Soft Electronics in Healthcare Wearables for Better User Compliance"

_micromachines, 2022, doi:10.3390/mi14010121_

Round 1

Reviewer 1 Report

Interesting work and helpful for getting knowledge for applications of smart sensors and material in  electronics. 

But more application papers can be cited in this paper as

1) IoT based assistive companion for hypersensitive individuals (ACHI) with autism spectrum disorder, https://www.sciencedirect.com/science/article/abs/pii/S1876201819306847

2) IoT Fog-Enabled Multi-Node Centralized Ecosystem for Real Time Screening and Monitoring of Health Information

https://www.mdpi.com/2076-3417/12/19/9845

3) Challenges, Applications, and Future of Wireless Sensors in Internet of Things: A Review

https://ieeexplore.ieee.org/document/9698203

Author Response

We thank the reviewer for the invaluable efforts and time to review our article. The mentioned articles are now added to the references list for IoT based sensors.

Reviewer 2 Report

1.       How do the authors determine the term “smart” in “smart materials” and “smart devices”? A clear definition is suggested to be given.

2.       There have been numerous review papers summarizing the state-of-the-art advances in soft electronics and wearable devices, and the majority focus on the materials science domain. So the uniqueness and necessity of the current submission is not clear which needs to be further strengthened.

3.       The title of Section 3.1 is missing.

4.       What are the physiological parameters commonly considered for wound monitoring? The authors are encouraged to supplement some details in this respect in Section 3.1.5.

5.       The authors need to clearly distinguish the different roles of smart materials in specific applications. Some are for sensing (e.g., chromic sensing of lactate), whilst the others are for treatment (e.g., stimulus-responsive wound dressing). For the current submission, they are confused.

6.       Microneedles are a powerful tool for accurate assessment of physiological conditions via wearable sensing of metabolites in interstitial fluid. The rapid advances of this field are advised to be supplemented to offer a comprehensive view to the readership. Here are some latest works to be considered:

Nat Biomed Eng, 2022, 6, 1214.

Adv Funct Mater, 2022, 32, 2112045.

Adv Sci, 2021, 8, 2103030.

Adv Mater, 2020, 32, 2002129.

Adv Funct Mater, 2020, 32, 2009850.

7.       Typos need to be carefully fixed. A typical example lies in the chaos of uppercases in Table 2 and some section titles.

Author Response

We thank the reviewers for their invaluable efforts to review our article. Response to the reviewers comments are provided in the attached file. 

Reviewer 3 Report

All the enlisted references are not cross cited into the manuscript, therefore it is inappropriate to say the authors have referred actually. The Self citation and Plagiarism slightly for reference [14]

In the given manuscript author(s) have done an good literature survey on the smart materials used in Healthcare Wearables for clinical applications. The authors have well organised written the article.

However, there are a few points that may improve the manuscript and will get more impact in this research area.

1) In the abstract, highlight the contribution of this article rather than a general description.

2) Throughout the article, cross-citation of the vast literature survey(refer to the list of references) is missing.

3) As per the title, in the section2, it would be better if a survey were done along with some commercial market device and their specifications(sensing material used) were tabulated. 

4)  Authors have defined the properties well in section 2 at line no. 110 to 112. Based on these properties, the existing wearable devices can be categorised and tabulated in review.

5) There must be a section on different smart materials, crucial parameters and their working principle. Ref. line 150 to 156.

6)  In Table 1 also add some developed manufacturers, with citations.

7) At line no 200, remove the spacing between the character.

8) Throughout the manuscript, mention the full form of abbreviations used at the first initialization.

9) In Figure 1,  the labels and text are unclear; rewrite FigA labels with FigB labels with different labels.

10) In Figure 2, Fig. 3, Fig. 4 and Fig. 5, improve the labels and describe the figure details with all shown information.

11) At line no. 300 to 303 shows the process either by the flow chart or the arrow marking.

12) In the table2, add up some commercial devices technology with cross-citation.

13) At line no. 366 to 368 qualities are highlighted in the need categories from the survey.

14) At self-repairing (line no. 375) or calibration needs to be described or tabulated according to the literature survey.

15) put one subsection on challenges of wearable devices enlist or tabulate them.

16) If possible cite all the enlisted reference in the article.

Author Response

We thank the reviewer for their time and effort to review our article. The response to their comments are provided in the attached file. 

Round 2

Reviewer 2 Report

The authors have made necessary revisions and the reviewer would suggest the manuscript to be accpeted at the current stage.